# Biodegradable Nonwoven Agrotextile and Films—A Review

**DOI:** 10.3390/polym14112272

**Published:** 2022-06-02

**Authors:** Dragana Kopitar, Paula Marasovic, Nikola Jugov, Ivana Schwarz

**Affiliations:** Department of Textile Design and Management, Faculty of Textile Technology, University of Zagreb, Prilaz baruna Filipovica 28a, 10 000 Zagreb, Croatia; paula.marasovic@ttf.unizg.hr (P.M.); nikola.jugov@ttf.unizg.hr (N.J.); ivana.schwarz@ttf.unizg.hr (I.S.)

**Keywords:** biodegradability, nonwoven fabric, mulch, natural fibres, biopolymers, recycled mulches

## Abstract

As society becomes more aware of environmental pollution, global warming, and environmental disasters, people are increasingly turning to sustainable materials and products. This includes agrotextiles in a wide range of products, including nonwoven agrotextiles for mulching. This review provides insight into relevant available data and information on the condition, possibilities, and trends of nonwoven mulches from natural fibres, biopolymers, and recycled sources. The basic definitions and differences between biodegradation and composting processes are explained, and the current standards related to biodegradation are presented. In addition, an insight into the biodegradation of various nonwoven mulches and films, including their advantages and disadvantages, is provided, to predict the future directions of nonwoven mulches development.

## 1. Introduction

The increasing concern for the prevention of ecocide, as well as the need for eco-friendly materials, has caused growing attention towards polymers made from sustainable natural sources and biodegradable plant materials [1]. Media attention and product marketing have led to a rise in consumer expectations regarding material sources, production, and disposal after usage. Additionally, stricter government regulations, regarding the product impacts on the environment, apply to a range of industries including agrotextile manufacturers [2].

Most agrotextile products are produced with synthetic materials based on petroleum, making them a typical artificial waste causing environmental pollution [3,4]. The global agrotextiles market size in 2020 was valued at USD 9.05 billion and is expected to grow at an annual growth rate of 4.7% from 2021 to 2028. Rising demand for agriculture products of better crop quality is expected to increase productivity over the forecast period and have a positive impact on market growth [5].

Respectively, inadequate and improper disposal of synthetic polymeric waste, due to the influence of ultraviolet radiation, physical abrasion, thermal oxidation, and microbial synthesis, cause decomposition in fragmented plastic particles, classified as microplastics [3,6,7,8,9]. Environmental contamination by microplastics is considered an emerging threat to biodiversity and ecosystem functioning. A significant amount of microplastics are stored in soil ecosystems (above and below ground), where their impacts on soil ecosystems remain largely unknown. Limited studies indicate that large amounts of fibrous and fragmentary microplastics are found in terrestrial ecosystems worldwide [10,11].

The accumulation of microplastics in the soil can result in adverse effects on crop production. It refers to the reduction in plant species and crop nutrient content, as well as the decrease in microorganism activity in the soil, which negatively affects crop growth [12,13]. Soil contamination by microplastics has a negative influence on soil structure, moisture, and nutrient transport in soil, causing root growth retardation and greenhouse gas emissions [14,15,16,17,18].

On the other hand, the rise of developing economies as well as an aging population is causing a further strain on food resources. By increasing the world population, the productivity of food production is growing, as well the demand for food quantity and quality. Although the productivity of food production increases, it causes an additional negative impact on the environment and represents a serious threat to the environment and future food production.

Solutions to overcome the stated problems could be found in sustainable agriculture. Sustainable agriculture is a system of plant and animal production practices, designed to maximize the quality and quantity of food for humans, protect and enhance the environment, conserve natural resources, maximize the use of resources, and sustain farm operations in the long run [19]. Sustainable development needs to ensure proper soil management, as a natural resource that cannot be replaced [20]. Sustainable development depends not just on soil health, but also on a system that is resource-conserving, socially inclusive, competitive, and environmentally friendly [21].

Contradictory to sustainable agriculture is the current usage of a huge quantity of plastic agriculture products. For instance, in 2019, about 63% of agroplastic, non-packaging waste generated in the EU was collected. The destiny of the remaining 37% is unknown, but it is assumed that was probably stored, burnt, buried, or collected with other waste through a waste stream. It should be noted that the collection rate varies among EU countries, where Ireland, Iceland, Norway, Sweden, France, and Spain have the highest collection rates of more than 70%. Although agroplastics have great potential for recycling, only 24% of agroplastics in the EU market are recycled. The percentage of agroplastics collected by type differs significantly, with no recycling reports for mulch and bale nets, while greenhouse film collection and recycling are well established. It is a worrying estimation that soil contamination in the EU is around 467 kilotons per year, with 166 kilotons (36%) accounted for mulch film collection. On the EU market, mulch films represent only 12% of agroplastic by mass [22,23].

Natural fibre-based agrotextiles, as one of the most popular alternatives fibre-based agrotextile group, have a thriving market in the agrotextile sector accounting for roughly 8.2% by volume and 6.4% by value of the worldwide technical textiles market in 2010. The global agrotextile market grew at a 3.9% annual rate from 1,615,000 tons (USD 6.5 billion) in 2005 to 1,958,000 tons (USD 8.1 billion) in 2010. According to a David Rigby Associates analysis, global end-use of agrotextiles climbed from 3.3% in 2000 to 3.9% in 2010 [24].

An overview of the current situation, trends, and shortcomings of biodegradable nonwoven mulches is given. The insight into the facts about raw materials and biodegradation of biopolymers, which potentially creates confusion, is provided. A review of the available literature on agrotextiles biodegradability and expected time of degradation for mulches made of natural raw materials, biopolymers, and recycled materials are given, to gain insight into future directions of biodegradable mulches development.

## 2. Biodegradation, Related Standards and Test Methods

Biodegradability is the ability of a polymer to degrade with help of living organisms into basic substances, such as water, carbon dioxide, methane, basic elements, biomass, and humic matter [2,25].

The degradation of the polymer chain leads to a decrease in the molecular weight of the polymer due to microorganisms acting under aerobic and anaerobic conditions through chemical reactions, such as oxidation, photodegradation, and hydrolysis, manifested by loss of physical, mechanical, and chemical properties [26].

Organic matter deteriorates by the action of microorganisms, bacteria, and/or fungi, during aerobic or anaerobic biological processes, depending on local conditions and the availability of oxygen. Moisture content, oxygen availability, temperature, type, and quantity of microorganisms (bacteria, fungi) and enzymes, as well as salt concentration, determine the degree and rate of biodegradation in different environments [27,28].

The difference between anaerobic and aerobic biodegradation models (Figure 1) [27,29] is in the presence (aerobic) or absence (anaerobic) of oxygen.

Aerobic biodegradation usually means composting under industrial and composting conditions. Microorganisms consume polymers as carbon and energy sources in an oxygen environment (not less than 6% oxygen), producing carbon dioxide and water as degradation products, as well as the compost residue, which is compost (Equation (1)) [27,28]. The material is oxidized into carbon dioxide (CO_2_), water (H_2_O), and some organic residue (non-degradable material and metabolites of microorganisms) and biomass (microbial carbon, i.e., carbon in microorganisms) [23]. Industrial composting is performed in a warm (approximately 60–70 °C) and moist (approximately 60%) environments under controlled conditions (pH 8.5) [27,28].
C_sample_ + O_2_ → CO_2_ + H_2_O + C_residual_ + C_biomass_(1)
Organic matter + S + O_2_ → CO_2_ + H_2_O + NO_2_ + SO_2_ + Heat + Compost

Anaerobic biodegradation typically involves anaerobic digestion in oxygen-free conditions (leading to the formation of methane gas, carbon dioxide, water, hydrogen sulphide, ammonia, and hydrogen, resulting in a sequence of metabolic interactions between various types of microbes) in mesophilic (37 °C) and thermophilic (55 °C) biogas plants. The remaining part is the digested residue (Equation (2)):C_sample_ → CH_4_ + CO_2_ + C_residual_ + C_biomass_(2)
Organic matter + H_2_O + Nutrients → Digestate residue + CO^2^ + CH^4^ + NH^3^ + H^2^S + Less heat

Depending on the environment, each polymer will have a typical degradation period; for example, cellulose can degrade in less than a year, while polyolefin can degrade over hundreds of years [2].

Synthetic materials, where polyethylene and polypropylene are cheap and the most widely used materials in agriculture, slowly degrade due to inactive groups, which are resistant to biological attacks [2].

Due to the slow degradation of plastic waste used in agriculture, during the decades, accumulation of waste occurs.

In order to control waste decomposition, light-sensitive antioxidants are applied to plastic materials, which help them to photodegrade into small fragments, leading to greater and irreversible pollution of agricultural land by microplastic.

The above-stated has driven many researchers to investigate agrotextiles that would retain their physical and mechanical properties during use, after which they will degrade into methane (anaerobic biodegradation) or CO_2_ (aerobic biodegradation), water, and biomass [31].

### Standards and Methods of Biodegradation

Due to limited performances and higher costs, biodegradable polymers are not commercially widely available. To determine their biodegradation in soil conditions, the level and time frame of degradation should be defined. The standard specifications NF U52-001 and UNI 11462, which refer to standard DIN EN 17033 (Plastics–Biodegradable mulch films for use in agriculture and horticulture–Requirements and test methods) are the only documents that specify biodegradation time frame and levels of biodegradable materials used in agriculture and horticulture [32].

The main requirements for mulching films are: (1) biodegradation of at least 90% within 24 months; and (2) material must not contain heavy metals, with no ecotoxicological effects [32]. In Table 1, the standards for biodegradability in the soil environment by measuring O_2_ consumption or CO_2_ production are given.

Previous studies show that PLA cannot be completely recycled since the biodegradation rate is 60% to 80%. On the other hand, cellulose-based biopolymers achieved a biodegradation rate of 70% after 350 days.

Limited biodegradation rates raise fundamental questions about whether materials classified as “biodegradable” are truly biodegradable or can only be composted.

According to standard ISO 14.855, which describes composting control, a material meets the biodegradability criteria if 90% of the initial mass is lost within 6 months at 59 °C. The additional provisions indicated in ASTM D5338 determine biodegradable biopolymer blends as biodegradable if their weight loss within 180 days is 90%. The acceptable rate of weight loss of homopolymers after 180 days is 60%. Material that does not meet the cited biodegradable criteria can be categorized as compostable. Compostable plastic is biodegradable, but biodegradable plastic cannot be composted; e.g., the material weight loss during the biodegradable test defines the difference between biodegradable and compostable material [33].

## 3. Biodegradable Nonwoven Agrotextile and Films

### 3.1. Agrotextile from Natural Fibres

The use of natural fibres for new materials development is one of the priorities of the Food and Agriculture Organization (FAO) as their use is expected to increase the efficiency and sustainability of agricultural industries worldwide.

Natural fibres are biodegradable, renewable, widely available, and have a neutral CO_2_ emission. Their properties include great mechanical strength, modulus and moisture absorption, along with low density, elongation, and elasticity. The production is relatively easy with low processing costs, which put them in a position of important economic focus for developing countries [31,34,35,36].

Plant fibres are widely classified according to their botanical origin as bast (kenaf, hemp, ramie, flax, jute, banana, etc.), leaf (sisal, agave, abaca, PALF, etc.), seed (coir, cotton, kapok, soya, rice hulls, etc.), fruit (coir, oil palm, etc.), stalk (rice, wheat, maize, rye, oats, etc.), grass (bamboo, bagasse, etc.), and wood fibres (softwood and hardwood) [37,38,39].

Plant fibres are mainly composed of cellulose, hemicelluloses, lignin, pectin, and wax, where the proportions vary depending on the type and part of the plant fibres [40]. Lignin is one of the main components that create a protective layer preventing the internal structure of fibres from degrading. Lignin is durable and not soluble in water, acting as a glue to cellulose and hemicellulose [41]. Variations in the biochemical composition of plant fibres depend on the type, age, and maturity, as well as soil type, climatic conditions, and methods of extraction and degradation [42].

Natural fibres used for agrotextile products provide environmentally friendly and promising opportunities [24]. The most commonly used natural plant fibres for agrotextile products are jute, coconut, sisal, flax, hemp, and wool [41]. Their application is for weed management and agro-mulching, sampling bags, baler twines, as a bed for seed germination, and agricultural product packaging (seeds, food grains, sugar, vegetables, and fertilizers) [24].

The microbiological degradation of cellulose occurs via enzymatic hydrolysis, which involves multistage hydrolysis of cellulose to glucose [43]. Cellulose provides fibres with good strength, stiffness, structural stability, and determines hygroscopic properties [44,45]. The plant structure is strengthened through the presence of pectin, an acidic heteropolysaccharide composed of modified glucuronic acid and rhamnose residues [46].

As opposed to chemical composition, natural fibres are highly variable in terms of their physical properties. Depending on the amount of cellulose in the fibre, the microfibrillar angle and the degree of cellulose polymerization, natural fibres manifest different mechanical properties [47]. The cellulose content of plant fibres influences their mechanical properties, where tensile strength and Young’s modulus increase by increasing cellulose content, but fibre strength and stiffness depend on hydrogen bonds and other chemical interactions in the cellulose [47,48].

Conventional natural fibres, such as cotton, wool, and linen, can be biodegraded if disposed on landfills or/and industrial composting. Cellulose fibres deteriorate quickly, in periods from 1 to 6 months, due to the presence of a glyosidic bond that is open to depolymerization by interaction with enzymes; the organisms consume the resulting glucose products. Generally, natural materials beside biodegradability have significantly lower impact on water and carbon dioxide emission providing desirable physical properties [2].

The disadvantage of natural fibres is their short lifetime comparing to synthetic materials, because of poor resistance to microbes and pests, high moisture sensitivity, and moderate thermal stability [2,48]. The environmental impact of natural materials refers to the cultivation and processing of natural fibres and not just to the end-of-life impact as it is for synthetic materials [2].

Plant and animal fibres, such as jute, coconut, or wool, may be a competitive alternative to synthetic materials for short lifespan agrotextile products [49]. Firstly, natural fibres are highly adaptable to modification as they contain several OH groups within cellulosic and lignin compounds [46]. By chemical treatment OH groups are modifying, where the fibre hydrophilicity can be reduced, while mechanical properties as well as moisture resistance can increase [50]. By biodegradation, natural fibres are decomposed into elements that do not represent a danger to the environment and could be used as a plant fertilizer. The price of natural fibres is lower compared to synthetic fibres, but further analysis of product development costs, field application, monitoring, and effectiveness evaluation is needed [47].

The physical and chemical properties of natural cellulose fibres along with the rate of biodegradation is presented in Table 2.

The composition of cellulose fibres directly affects the rate and degree of biodegradation. The structure of a fibre is determined by the monomer shape, structure, and configuration, orientation of the macromolecules, and the bonds between them, as well as a degree of polymerization (DP). The primary and secondary properties of the fibre are determined by the ratio of crystalline and amorphous areas of the fibre. Primary fibre properties, such as length, fineness, strength, flexibility, cohesion, and uniformity, along with secondary properties, such as surface micro characteristics and cross-sectional shape, fibre crimp, moisture absorption, and fibre density, determine the initial process of biodegradation. Cellulose is a semi-crystalline polysaccharide with a large number of hydroxyl groups. The result is a very poor interface and low resistance to moisture absorption. Hemicellulose is strongly bound to cellulose fibrils by hydrogen bonds. Because of its open structure, which contains many hydroxyl and acetyl groups, hemicellulose is partially water-soluble and hygroscopic. Lignin and pectin act mainly as binders. Lignins are amorphous, highly complex, mainly aromatic polymers of phenylpropane units, with the lowest water absorption of all natural fibre constituents. Lignin binds hemicellulose and cellulose preventing cellulose fibres from swelling in water. The waxy substances in the fibres affect the wettability of the fibres and the adhesion properties. If the composition of cellulose fibres is considered, it is obvious that biodegradation time is not determined only by the amount of cellulose. For example, the biodegradation time of bamboo fibers varies (to 36 months) since it depends on the age of the plant and the method of fiber extraction. Studies indicate that lignin has a major impact on the mechanical properties of extracted fibers. The extraction of bamboo fibers can be performed mechanically (steam explosion, soaking, crushing, grinding, and rolling), chemically (degumming, alkaline or acid cooling, and chemical cooling), and by combined mechanical and chemical extraction. In general, the chemical content of bamboo changes with the age of the bamboo plant. In particular, the cellulose content decreases with the age of bamboo, which directly affects the chemical composition of bamboo fiber. The main components of the secondary cell wall of bamboo are cellulose, hemicellulose, and lignin, which affect the chemical composition of the fibers. Abdul Khalil et al., reported that the cellulose content decreases continuously with the age of bamboo, which affects the chemical composition of the fibre. Various treatments cannot completely remove the lignin content of bamboo fibers as lignin has proven to be quite resistant to various alkalis. Lignification of bamboo is a dynamic process of lignin deposition in the secondary cell wall, which depends on the ratio of monomers S, G, and H in the lignin structure. Studies indicate that the weight of bamboo lignin increases with age and that the ratio of S to G monomers increases. The percentage of hemicellulose directly affects the ability of moisture absorption. The cellulose fibres with high hemicellulose percentage easily absorb moisture leading to fibre swelling, consequently allowing microorganisms to enter the fibre structure where the biodegradation process begins. On the other hand, lignin binds cellulose and hemicellulose together, preventing moisture penetration and swelling of the fibre, respectively, providing a fibre barrier to microorganisms that would initiate the biodegradation process [49,51,62,63].

### 3.2. Mulches from Natural Fibres

The benefit of mulching is its ability to reduce evaporation, increase moisture retention, regulate temperature, enhance nutrient availability, and root absorption, suppress weeds, decrease salinity, promote biological activity, control crop pests and diseases [64,65,66,67,68,69,70,71,72,73,74,75,76,77,78].

The use of organic mulch materials is common for agriculture land through crop rotation systems in order to improve soil health, where, in the last few decades, the use of inorganic plastic mulch gained popularity [79]. The extensive use of inorganic plastic mulch leads to severe environmental pollution, with a negative impact on soil quality, resulting in the reduction in agricultural productivity and a threat to sustainable development [21,66]. For sustainable development is necessary to monitor and evaluate the health of agricultural soil related to different mulching materials, as well as local environmental conditions. In general, inorganic mulching materials, such as plastic films (foils), should be chosen in accordance with conditions in the local area and farmers’ specific needs, while organic mulches can be a viable and sustainable alternative that, in addition, improves soil health [21,67]. The most effective solution for plastic mulch pollution is to recycle plastic films or use biodegradable mulches (Figure 2). Mulching can be broadly categorized as organic or inorganic, depending on the type of materials that are used, which depends on availability, cost, decomposition rate, durability, effect on soil properties, and functions [80].

Organic mulches are an attractive option for growers due to the possibility of agricultural recycling on farms, where their decomposition during the growing season can serve as a source of nutrients without creating any waste disposal costs at the end-of-life cycle [81].

Since weeds, like any other plant, need sunlight for their own growth and development through the process of photosynthesis, the use of mulches controls weeds grow in a natural way without the need for the chemicals usage as shown on Figure 3 [81]. In periods of increased precipitation, organic mulch, due to its permeability, increases soil moisture as opposed to impermeable plastic mulch. Furthermore, mulching can improve early yields, crop yields, and efficiency of water usage. Unfortunately, below organic mulches, soil temperature could be reduced, which can lead to reduced crop yields [82].

Arshad in his research investigated the time span of cellulosic materials (cotton fibres by an average diameter of 17.16 µm, jute fibres of 68.00 µm, linen of 24.38 µm, flax of 70.64 µm) and wool with an average fibre diameter of 23.04 µm. The mass per unit area of the materials was 182 g m^−2^ for cotton, 263 g m^−2^ for jute, 211 g m^−2^ for linen, 413 g m^−2^ for flax, and 198 g m^−2^ for wool fabrics [79]. The biodegradation process of materials proceeds in a similar way with a difference in biodegradation time. Microscopic observation, as well as FTIR and TGA analysis, indicated that the major portion of cellulose has been degraded by microorganisms. Wool showed better resistance to the microorganisms due to its molecular structure and its surface.

A group of scientists described the biodegradation of light nonwoven cotton fabric (50 g m^−2^) by zero-order biodegradation kinetics in aerobic soil. High water pressures during the hydroentangling process of light cotton nonwoven fabric causes the partial removal of waxes and retained pectin of the cuticle, as well as surface fibrillation, which affects the biodegradation rate [83]. Since the cuticle is destroyed by the high water pressures of water jets during the bonding process of nonwoven fabric, bacteria, and fungi easily break down cellulose by the action of their enzymes.

Investigation of cellulose fabrics biodegradability by soil burial test, an activated sewage sludge test and enzyme hydrolysis showed that linen fabrics (250 g m^−2^ and 0.63 mm thickness) are the most biodegradable, followed by rayon (75 g m^−2^ and 0.19 mm thickness), cotton (100 g m^−2^ and 0.29 mm thickness), and, finally, acetate fabrics (83 g m^−2^ and 0.13 mm thickness). Small animals like earthworms were found only in the soil where linen fabrics were buried, which is explained by the largest portion of non-cellulose ingredients of linen fibres, making it a favourable place to break into the internal structure for annelids, arthropods, as well as microorganisms resident in soil [84].

Among cellulose-based materials, viscose is becoming popular for mulching and production of plant seedlings due to its good sorption properties and fast biodegradation. Biodegradation time depends on many factors and can range from a few to 10 weeks. The viscose nonwoven fabric of 126 g m^−2^ modified by padding with KNO_3_ and then coating with PLA, degrades in 15 weeks, providing the cultivated tomato plants optimal thermal and water conditions, as well as protection from weed infestation. Initially, before the biodegradation process, air permeability was 2758 dm^−3^ m^−2^ s^−1^, the perpendicular water permeability 72 × 10^−3^ m s^−1^, breaking force and elongation were 11.48 N and 7.78 mm. After 14 weeks of exposure, air permeability (2965 dm^−3^ m^−2^ s^−1^) and perpendicular water permeability (178 × 10^−3^ m s^−1)^ increase, while breaking force decreases to 1.87 N. After 14 weeks of study, the nonwoven sample was quite degraded and the elongation took the value of 10 mm [85].

In research on bast fibre biodegradability, mulches from biodegradable bast fibres (flax and hemp) are produced by a mechanical process where the web is bonded by pre-needling and a hydroentanglement process, forming nonwoven fabrics of 200 g m^−2^ and 0.5 mm thickness; they were compared to commercially black PE split film woven fabric. Hemp and flax fibres contain less than 5% shives. Retted flax fibres had greater length, finer diameter, and darker colour than unscavenged hemp fibres. During a weed assessment study, conducted over a period of 31 to 88 days, the nonwoven mulch produced from hemp fibres treated with carbon black-based pigment improve soil heating efficiency. Additionally, preliminary research has shown that dyeing bast fibres of nonwoven fabric in black achieves the same weed suppression efficiency as black polyethylene woven mulch. Degradation of the nonwoven fabric by worms and insects led to material decomposition to the extent that they could be easily cut with a garden spade. This is particularly important in the usage of the fabrics on a large scale to allow agricultural machinery to break down the material [81].

In their research, Zhou and associates [86] used air-laid nonwoven fabric made of 86% waste fibres from the textile industry (ramie and cotton fibres with a mass ratio of 4:1) bonded by modified corn starch aqueous solution (14%) in a drying chamber, for the biodegradability test. The ramie fibres were 2 to 5 cm in length, diameter of 30 µm, density of 1.49 g cm^−3^, and 6.5% of moisture content. The cotton fibres were about 13 mm in length, 20 µm in diameter, density of 1.58 g cm^−3^, and with 7.2% moisture content. Biodegradation began rapidly and intensively within 4 days, achieving an average degradation rate of 4.08% per day. For comparison, control reference cellulosic material had an average rate of 4.73% per day. Accelerated degradation of nonwoven fabric is the result of modified corn starch which degraded more easily than ramie fibres. By day 60, ramie/starch nonwoven fabric degraded by 64.5%, with an approximately constant average degradation rate of 0.86% per day. The reference cellulosic material had a significantly higher degree of degradation, where at the end of biodegradability period (day 72), the degree of degradation of the reference cellulose material and nonwoven fabric were 77.1% and 65.6%, respectively.

In the case study by Sanjoy Debnath, nonwoven Geojute is considered to provide a microclimate necessary for the initial vegetation of seedlings, giving the soil N, P, K, and other minerals during its own biodegradation. The investigation includes woven fabric of 150 g m^−2^ for seedlings and nonwoven jute fabrics of 150, 200, and 250 g m^−2^ for mulching. The rate of degradation of jute nonwoven fabric is faster than jute woven fabrics, which leads to faster growth of seedlings in the vertical direction and coverage in the horizontal direction. The strength of the nonwoven fabrics was greater in cross machine direction than in the machine direction. The nonwoven fabrics have poor initial and secant modulus, but field testing did not reveal a major impact on the fabric’s performance when used for mulching [87].

Sampling pots made of different woven and nonwoven jute fabrics weight, reinforced by different modified soy percentages were tested on biodegradability [88]. Different sets of jute felts in a combination of 40, 50, 60, 70, and 80 wt.%, were soaked by immersion in the corresponding sets of modified soy resin and then partially dried. The jute felt was pressed with a hydraulic press to obtain the nonwoven jute soy modified composites. For the biodegradability test comparison, PE bag was used. After 60 and 120 days, depending on jute fabrics weight and modified soy percentage, the sampling pot walls were loosened and mostly degraded. By its own degradation, the fertility of the soil increased as jute and soy resins are made up of carbon, hydrogen, and oxygen. In addition, planting can be made directly in the soil allowing easy propagation of plant roots through jute pots, which is not possible in the case of PE or cement/earthen pots.

In research conducted by Mańkowski et al., the biodegradability of barrier nonwoven sanitary mats, with mass per unit area of 900 and 1350 g m^−2^, composed of three layers, was tested. The surface layer was composed of 80% hemp and 20% flax fibres, and the middle layer, composed of jute, was joined together by stitch needling using natural fibres to reinforce the nonwoven structures. The third layer consists of flexible natural resin (latex). The tensile strength of nonwoven sanitary mat, with mass per unit area of 900 g m^−2^, dropped from 140.7 N to 78.3 N after 4 months, 14.3 N after 8 months, and finally 90% after 12 months of exposure. Reduction in tensile properties is also present in samples of nonwoven sanitary mats, with a mass per unit area of 1350 g m^−2^, from the initial 160.2 N, to 93.2 N after 4 months, 14.7 after 8 months, and almost complete reduction after 12 months of exposure [89].

The studies confirm that 90% of nonwoven fabrics biodeterioration after 10 months based on sensory evaluation and tensile properties analysis, confirming the high vulnerability of flax/hemp fibres to biodegradation.

## 4. Biodegradable Nonwoven Agrotextile and Films from Cellulose Regenerates and Biopolymers

### 4.1. Cellulose Regenerates and Biopolymers in Agrotextile

There is a growing trend to produce biopolymers on a large scale, for a variety of applications, with prediction they may replace petroleum-based polymers between 30% and 90% by the year 2050 [90]. Most of synthetic fibres have a low rate of biodegradation and remain in the environment for a long time. This applies in particular to high-performance nonwovens made of stabilized polymers, which are highly resistant to physical, chemical and biological attacks typical in landfills environment. The biodegradation rate of synthetic materials, when buried in soil, ranges from almost non-biodegradable materials (such as PET and PE), through materials that degrade for 95% in 6 weeks in typical landfill conditions (such as PCL and PLA), to materials that reach the same percentage of biodegradation in just 8 days (Lyocell) [81].

In recent decades, many studies have been conducted to develop and industrialize so-called biodegradable plastics that would not accumulate in the environment. An example is oxo-degradable plastic, which is essentially conventional plastic (e.g., PE, PP, PET) with additives (prodegradants) that accelerate the oxidation process. The problem is only partially solved since the oxo-degradable plastic used in mulching quickly deteriorates to fragments, while their complete biodegradation takes a long time [91]. Second, the fragments of plastic mulch can adsorb pesticides and fertilizers from the soil, resulting in deeper soil and water pollution. Portillo et al. and Feuilloley et al. found that the degradation rate of photo-degradable PE and oxo-degradable PE film does not reach the requirements of current international standards [80]. The biodegradability of biodegradable plastics, such as PLA, PCL, and PBAT, depends on the polymer properties, additives incorporated into the final product, as well as the environmental conditions in which the material ends up [91].

A variety of novel biomaterials has been developed by modifying biopolymers or by synthesizing bio-inspired macromolecules. Designs of such intelligent biomaterials are green and environmentally friendly [92]. Almost all biopolymers are biodegradable and can be microbiologically decomposed into carbon dioxide (CO_2_), water (H_2_O), methane (CH_4_), and inorganic compounds [93]. Along with a biopolymer type and chemical composition, environmental conditions affect its degradation capacity [36].

Biopolymers are polymers produced by living organisms. Based on their origin, three types of biopolymers can be distinguished as natural, synthetic, and microbial biopolymers [90]. According to their biodegradability, biopolymers can be classified as biodegradable and non-biodegradable. Alternatively, biopolymers can be divided into bio-based and non-bio-based biopolymers, where some biopolymers can be made from both bio-based and fuel-based resources (PLA, PBS, PTT). Based on their response to heat, biopolymers can be divided on elastomers, thermoplastics, and thermosets. Summarized, biopolymers can be classified differently based on a different scale [94].

If the division of biopolymers based on their origin is taken into account, natural biopolymers can be found in animals (hydrocarbons, proteins, fats, nucleic acids, etc.) and plants (e.g., cellulose, oils, starches, even polyesters). Natural polymers are formed in cells of living organisms in the cycle where the cell is growing. Microbial polymers are produced by several microorganisms, where the polymer is attached to the cell surface inside the microorganism. The polymer is synthesized by enzyme-catalysed polymerisation of activated monomers, which occur within cells as products of metabolic processes. Materials created by nature are degraded by nature using enzymatic systems inside the cells. During times of abundance the organism synthesizes polymers (biopolymers formation), and in times of scarcity it consumes them (biopolymers degradation). Bacteria use these microbial biopolymers as storage materials in response to particular environmental stresses.

Biopolymers of artificial origin are polymers usually derived from oil, e.g., they do not occur in nature, so when they reach the natural environment they cannot be incorporated into the natural cycle by degradation. Degradability of those polymers is achieved with the integration of hydrolytically unstable bonds into the polymer (e.g., ester-, amide groups, etc.). Another source of artificial biopolymers is considered polymers produced in a manner identical to the natural but on an industrial scale. For example, many microorganisms in nature synthesise polyester as a substance for energy storage. By fermentation of sugar (glucose) under the influence of microorganisms and under the optimal industrial conditions, large quantities of polyester can be produced. Polyester is, therefore, a natural polymer but its production is closely controlled, therefore its source is considered artificial.

Bioplastics also belong to the category of biopolymers, where bioplastics can be made from natural, artificial, and microbial origins. Based on the raw materials used during production, biodegradable plastics can be divided into five different categories [95]:Starch based biodegradable plastics;Cellulose based biodegradable plastics;Biodegradable plastics obtained via chemical synthesis;Biodegradable plastics produced by bacteria;Biodegradable plastics of petrochemical origin.

In addition to division based on raw material, bioplastic can be divided according to the ability of degradation as biodegradable plastics (including compostable plastics) and plastics that are not biodegradable. The definition used today in industry denotes bioplastics as biodegradable plastics and/or plastics from renewable resources. It should be aware that biodegradable plastics made from renewable raw materials are not automatically biodegradable, while on the other hand biodegradable plastics are not necessarily made from renewable raw materials [95]. The term “bioplastics” often creates confusion about its concept. It is commonly incorrect to believe that if the material is obtained from biomass, then it must be biodegradable. It does not necessarily mean that the use of bio-raw materials must results in a biodegradable material, as well as that biodegradable plastics are always biologically based [96]. Currently, most of the certified biodegradable plastics are produced from renewable raw materials. However, there are also fossil-based biodegradable plastics that fulfil the requirements of different composability standards. Changes in functional groups, cross-linking density and copolymerization of non-biodegradable monomers may result in materials that can lead to biodegradable plastic, as well as combination of renewable and non-renewable raw materials [80,95].

Polyhydroxyalkanoates (PHA), poly (lactic acid) (PLA), poly (butylene succinate) (PBS), polyethylene (PE), poly (trimethylene terephthalate) (PTT), and poly (p-phenylene) (PPP) are the polymers containing at least one monomer synthesized via bacterial transformation. Among them, PLA, PHA, and PBS are well known for their biodegradability (Table 3).

Currently, the main representative of a synthetic biodegradable, renewable, and recyclable biopolymer is PLA. PLA can be produced by chemical synthesis or enzymatic polymerization. The chemical synthesis requires elevated temperatures, extremely pure monomers, and anhydrous conditions to avoid side reactions. The process uses heavy metals for catalysts where the heavy metal trace residues are undesirable for some applications (biomedical and food). Enzymatic polymerization is an environmentally friendly alternative for polymer synthesis since it can produce a fine polymer structure from low-cost raw materials (sugar beet, corn starch, sugar cane), making it a more desirable process than chemical synthesis. The polymer synthesis starts with LA production by microbial fermentation, then lactide formation and ends with LA ring-opening polymerization [88,97]. Today’s studies are focused to discover natural PLA producing microbes and accomplishing the one-step microbial production of PLA. The biopolymer is hydrophobic, biocompatible, biodegradable, and thermoplastic and has unique characteristics, such as strong clarity and high rigidity [47,93,97,98].

PHAs are a family of intracellular biopolymers, e.g., polymers that are synthesized via various Gram-positive and Gram-negative bacteria as intracellular carbon and energy storage granules in nutrient-limiting conditions. These biopolymers are linear polyesters produced in nature by bacterial fermentation of sugar or lipids. PHAs composed of hydroxyalkanoate (HA) units, arranged in a basic structure that is obtained through bacterial fermentation. More than 150 different monomers can be combined within this family to provide materials with extremely different properties. The poly (hydroxybutyrate-cohydroxyvalerate) (PHBV) and poly (hydroxybutyrate-cohydroxyvalerate) (PHB) are examples of PHAs and have similar mechanical properties to PE and PP [99].

Polyvinyl alcohol (PVA), polyglycolic acid (PGA), polybutylene succinate (PBS), poly (butylene adipate-co-terephthalate) (PBAT), and polycaprolactone (PCL) are biodegradable polymers derived from petrochemical resources [99].

Polyvinyl alcohol (PVA) is a linear synthetic polymer produced by hydrolysis of polyvinyl acetate in order to remove the acetate groups. Degree of hydroxylation, number of OH groups in a molecule, influence the physical characteristics, and chemical and mechanical properties of the PVA. The PVA polymer is highly soluble in water, resistant to most organic solvents, mechanically strong and biocompatible [90,94].

Polybutylene succinate (PBS) could be one of the alternatives to the traditional non-degradable polymers due to its biodegradability in multiple environments. It can be produced from renewable or petroleum resources and been used in a wide range of applications, such as agriculture mulch, vegetation nets, compost bags, packaging materials, and others [100]. PBS is made up of 1,4-butanediol, and succinic acid, and can be easily processed as semi-crystalline thermoplastic polyester with acceptable thermo-mechanical properties comparable to PP [97,99]. Due to poor tensile properties, low melt viscosity and gas barrier properties in the polymer matrix, fillers such as clay, graphene, and carbon nanotubes could be incorporated.

Poly (butylene adipate-co-terephthalate) or PBAT is an aliphatic–aromatic biodegradable copolyester polymer produced from fossil sources. PBAT is a copolymer of butylene adipate and terephthalate made by melting poly-condensation [99]. The aromatic crystalline fraction of BT structure provides excellent physical properties, while BA non-crystalline structure of aliphatic chains degrades faster in several conditions [102]. The soil biodegradation of PBAT can be regarded as hydrolysis under the effect of microbial enzymes. Due to the flexibility and high elongation at break, PBAT is widely used as biodegradable blown film products mostly for the disposable products that degrade quickly enough to meet the home compostability criteria.

PCL is a hydrophobic and partially crystalline synthetic aliphatic polyester made by ring-opening polymerization of ε-caprolactone [99]. The polymer’s relatively low strength narrows the field of application in biomedical or tissue applications, the food-packaging industry and agriculture due to great resistance to water and oil, non-toxicity and biodegradability. Many attempts have been made to combine PCL with other polymers to modify its properties and degradation in the environment [103].

Dissolution of natural cellulose fibres can be achieved by derivative and non-derivative solvent systems. Cellulose is difficult to process because this natural polymer does not melt or dissolve in common solvents due to its inter- and intra-hydrogen bonds and its partially crystalline structure [104]. Viscose fibres are made by dissolving cellulose, which is then impregnated, breaking the hydrogen bonds between the macromolecules, increasing the molecular distance, and turning the fibre into alkali cellulose. The dissolution of cellulose usually takes place by grafting groups onto molecules to form a new intermediate in the dissolution of the derivative [105]. The raw material for cellulose dissolution is mainly wood and cotton linter. For the production of viscose fibre, the derivative dissolution takes place in sodium hydroxide or carbon disulphide, while the esterification process takes place in cellulose acetate fibre. Natural cellulose is exposed to mercerization, aging and xanthation treated with CS_2_. The produced cellulose xanthate dissolves in a dilute aqueous sodium hydroxide solution. The hydrogen bonds between the macromolecules are broken, the distance between the molecules is increased, and the fibres become alkali cellulose. During the aging process, catalysts play an important role in reducing the degree of polymerization of alkali cellulose as time and temperature change. The yellowing process causes cellulose molecules to combine with sulfonic acid groups to form larger molecules. By increasing the hydrophilicity of the molecules, the hydrogen bonding is further undermined and the cellulose is effectively dissolved. Then, the spinneret sprays the viscose into the coagulation bath to fix it into filaments. The coagulation bath usually contains sulphuric acid compounds [103,104]. In the production of cellulose acetate fibres, the cleaned cellulose fibre is treated with a mixture of acetic acid, acetic anhydride, and concentrated sulfuric acid for acetylation. After aging, the cellulose acetate flakes are precipitated and then dissolved in acetone [104].

### 4.2. Nonwoven Mulches and Films Made of Cellulose Regenerates, Biopolymers, and Their Blends

Gabrysa et al., modified commercial viscose nonwoven fabrics by dyeing, adding PLA layer on the surface and adding potassium nitrate (KNO_3_) giving mulches fertilizing properties in order to produce modern mulching material in agriculture. The weight loss of the mulches was 74% in 12 weeks and 88% in 14 weeks, and the tests were completed in the 15th week. The proposed modification of viscose nonwoven mulches can be successfully used as a multi-purpose and biodegradable nonwoven crop cover, eliminating the problem of post-harvest mulches disposal [85].

Steinmetz et al. in their research tested four nonwoven fabrics, with a mass per unit area of 50 g m^−2^ (raw cotton, viscose rayon, PLA, and PP), for biodegradation time. The best fit kinetic model to describe the biodegradation rates of raw cotton and rayon nonwoven fabrics was determined to be the first-order kinetics model:w_t_ = w_0_ e^−kt^(3)
where w_t_ is the weight of fabric remaining at a given time, w_0_ is the initial weight of the fabric, t is the time in days (d), and k is the first-order rate constant (d−1). Straight-line models were fit for the natural logarithm of the fabric weight remaining as a function of time. The initial model fits the fabric types simultaneously, allowing for the intercepts and slopes to differ by fabric type using analysis of covariance techniques. Due to low crystallinity and high moisture absorption capacity of rayon fibres, rayon nonwoven fabric lost more than 80% of the original weight within 28 days of exposure to soil. The rayon nonwoven fabric after 7 days of exposure exceeded the weight loss of raw cotton. After 63 days of research, no samples of raw cotton and rayon were observed in the soil. In contrast, PLA and PP nonwoven fabrics remained intact even after 140 days of exposure due to poor biodegradability of PLA materials and high resistance of PP fabrics to microorganisms [66].

Biodegradation evaluation of cotton, jute, linen, wool, viscose, polyester, and polylactic acid nonwoven fabrics was carried out by visual observation, weight loss, characterization of a chemical structure and surface morphology. A large part of biodegradation takes place in cotton and regenerated cellulose fibres, while synthetic fibres remained mostly undamaged at intervals of 1 to 4 months of burial test, concluding that cellulosic materials show the ability of disposal after use by burial in soil [106].

Currently, intensive research on the production technology, properties and applications of films and nonwovens made from biodegradable PLA, naturally occurring polysaccharides, thermoset polymers from vegetable oils and synthetic polymers with additives responsible for photo-, oxo- or biodegradation, are conducted (Figure 4). The addition of zinc, iron, cobalt, manganese, and magnesium accelerates long-chain polymers oxidation and degradation under the influence of heat, air, and light. The polymers with presence of free oxygen (aerobic environments) can break down by microorganisms into carbon dioxide or methane, water, biomass, and other organic compounds. The composition and concentration of additives that activate the shortening of polymer chains, for example, iron and calcium stearates are considered as great importance. Their presence in polymers leads to accelerated aging and deterioration of physical and mechanical properties [107]. The PLA polymer advantages are renewable agricultural sources (plants biomass), production with significant energy savings, and carbon dioxide gas exhaustion during production related to a synthetic fibres, recycling, and composting suitability (under the right conditions) and possible modification in order to obtain desired physical and mechanical properties comparable to synthetic polymers [1,31,108]. Polylactic fibre has proven to be an alternative to polypropylene (PP), polyethylene (PE), or polyethylene terephthalate (PET) because its properties are similar to non-biodegradable synthetic fibres in terms of controlled crimp, smooth surface, and low moisture absorption. PLA fibres have better weather resistance and flame retardancy relative to PET fibres, as well as rapid degradation in activated sludge. Its specific gravity, glass transition temperature, and modulus are within the values reported for polyamide (PA) and polyethylene terephthalate (PET) fibres [31]. Due to the complicated production, PLA is more expensive than other polymers, but as stated before, different manufacturing technologies are being developed to simplify the process [109].

The degradation of PLA under composting conditions by chemical hydrolysis is well known, but the degradation of PLA by microorganisms has not been completely investigated. The most adopted explanation is a two-step degradation mechanism including chemical hydrolysis at elevated temperatures followed by microbial degradation. At microbial degradation, microorganisms mineralize the polymer degradation products to produce carbon dioxide (aerobic conditions) or methane (anaerobic conditions) [110].

Investigation of microbe’s role in PLA degradation obtained in compost and in microorganisms rich soil showed that the fastest degradation occurs at elevated temperatures (45 °C and 50 °C), while at slightly lower temperatures (25 °C and 37 °C) the rate of degradation was low or almost non-existent during 12 months of exposure. The authors concluded that additional research to identify the microbes responsible for the enhanced degradation of PLA is needed [106]. A similar conclusion was presented in a study by Puchalski et.al. where PLA spunbonded mulching nonwoven of 50 g m^−2^ degraded after several weeks at higher temperature. The temperature must be near or above the glass transition temperature, which for PLA is in the range of 55–65 °C [111].

The PLA polymer is a good replacement for conventional synthetic nonwoven fabric used for vegetable growth since it transmits only 8% less radiation than PP nonwoven fabric. The investigation of cucumber yield in fields covered with PLA degradable and conventional oil-based nonwoven fabric shows the same yield [109,112]. Slightly higher temperatures under the PP and photodegraded PP mulches were observed and compared to control field. However, under the PLA degradable nonwoven fabric, lower temperatures than the bare soil were recorded.

Mulching in the vegetable field with photodegraded PP increased soil macroaggregate contents, while under PLA and Bionolle biofilms, smaller aggregates were noted. Photodegraded PP and PLA mulches provide a great increase in cucumber yield [113,114]. Furthermore, Bionolle biodegradable nonwoven covers (PBS), with a mass of 59 g m^−2^ and 100 g m^−2^, used to protect leek crops in the winter, provide higher leek yields [112].

Dharmalingam et al. investigated the degradation time of biodegradable bio-based spunbond black and white, as well as meltblown nonwoven mulches made of PLA and blend of PLA/PHA (70/30). Samples were tested in compost-enriched soil under controlled ambient conditions in a greenhouse for 45 days. The mass per unit area of mulches were 82.8, 79.1, and 84.2 g m^−2^, while thickness ranged from 593 to 666 μm. The initial tensile strength of spunbond black PLA nonwoven mulch was 37.64 N, black meltblown PLA 12.68 N and black meltblown PLA/PHA 3.13 N. After 45 days of exposure the tensile strength of spunbond black PLA nonwoven mulch decreased to 32.74 N. The tensile strength of black meltblown PLA and black meltblown PLA/PHA nonwoven mulches decreased to 1.75 N and 0.62 N, respectively; after 45 days of exposure tensile strength was not measurable due to significant degradation for both meltblown samples. The results suggest that during the first weeks of testing, microorganisms begin to utilize the more available components of the mulches as carbon sources, that is microbial activity is greater and occurs more strongly for the PLA/PHA blend [115].

Habolt et al. investigated similar mulches, spunbond black and white PLA, meltblown nonwoven PLA, and blend of PLA/PHA (75/25), tested throughout simulated weathering. Spunbond black and white PLA nonwoven have higher tensile strength than meltblown PLA and PLA/PHA nonwovens (56.21 N, 37.12 N, 8.96 N, and 3.90 N). The mass per unit area of the meltblown nonwoven mulches ranged from 75.6 to 85.4 g m^−2^ with the corresponding thickness ranging from 570 to 743 μm. The fibre diameter amount 7.3 μm for meltblown PLA mulch and 15.8 for spunbond black PLA. In the 21 days of the simulated weathering cycle, tensile strength values decreased (47.61 N, 4.12 N, and 0.21 N), except for spunbond black PLA where tensile strength increased from 37.12 N to 39.60 N. Both meltblown nonwoven mulches degrade more than 90% after 90 days and could be recommended for “Class II” material according to standard ASTM WK 29802 (the standard specification for biodegradability of agricultural plastics in soil). Spunbond nonwoven mulches met the standard compostability criteria of more than 60% biodegradation after 90 days (ASTM D6400) and they are more applicable for multi-season mulching and long-term agricultural applications [116].

The PLA nonwoven agrotextiles with different ranges of crystallinity (11.1–33.6%), under laboratory conditions that simulate natural aerobic conditions during composting, were completely degraded over 16 weeks. The influence of the degree of crystallinity on the degradation process dynamics was not recorded [117].

Although the biodegradable plastic mulches have existed in the market for more than 15 years, studies related to their impact on soil properties (aggregate stability, infiltration, soil pH, electrical conductivity, nitrate-N, exchangeable potassium), soil health indicators (hydraulic, biological, fertility, salinity, and sodicity), and soil functions (nutrient cycling) are limited. A two year evaluation period of soil health under influence of biodegradable plastic agrotextiles (two commercially developed materials made of polyester/starch blends, one commercially developed polyester mulch, complexing film of starch with polyesters, experimental product made of PLA/PHA) revealed that effects of biodegradable plastic mulch on soil health were mostly positive, although these effects were not consistent among the different mulch types, sampling times and site, indicating that additional research should be made [118].

In the field test by Wortman et. al., bioplastic films (Eco Film and Bio Telo) and four experimental spunbond, nonwoven biofabrics (3M Company, St. Paul, MN, USA), varying by thickness, weight and colour were investigated. The samples remained fully intact through the entire growing season of tomato and bell pepper leading to increased soil moisture and the elimination of weed competition. Biofabric mulches did not alter soil temperature relative to bare soil, whereas bioplastic films increased soil temperatures in the range of 1.7 °C to 2.3 °C. Increased soil temperature is often a desirable effect of agricultural mulches, but it can also contribute to physiological stress in warm climates and root disease in saturated soils [82].

The degradation of conventional PP nonwoven fabric and PP nonwoven fabric with an iron stearate photodegradation activator (PP photod. 0.1%) during cultivation of zucchini was performed. At the end of the zucchini vegetation, PP nonwoven fabric with a photoactivator showed a weight loss of 47.5%, while conventional PP nonwoven fabric mass per unit area increased due to the presence of soil particles on the surface of the fabric. Both fabrics have a large decrease in tensile properties, where a significantly largest decrease in the breaking force and elongation was recorded at PP nonwoven fabric with photoactivator [119].

Available literature indicates a positive effect of mulches on the yield and biological value of vegetable crops. The influence of conventional PP and PLA mulches on L-ascorbic acid, dry matter, soluble sugar, and nitrates content in tomato showed that PP mulch provides higher content of L-ascorbic acid in the fruit, while PLA mulch gives higher content of soluble sugars and dry matter, as well as a smaller concentration of nitrate ions related to the control field [120].

Since plastic mulching films are mostly used in strawberry growing, research to compare conventional PE plastic film with biodegradable mulches, under real and controlled conditions in a laboratory has been conducted. Biodegradable mulches were made of Mater-Bi polymer (a blend of PBAT and starch), as well as a combination of Mater-Bi with recycled material. Biodegradation was assessed by a respirometry test under laboratory conditions and in the field measuring the loss of mass per unit area over time. The study reveals that the great influence of weather conditions on mulch biodegradation. Precipitation was present only in the first two months of the experiment, when biodegradation may occur, and the dry period thereafter stopped the biodegradation process, resulting in a low value of mulches biodegradation. It can be concluded that biodegradation under real conditions is sometimes not consistent with biodegradation under controlled conditions. Due to environmental conditions, mulches biodegradation does not necessarily comply with bio-degradation standards, while this may be the case under controlled conditions. Compared to PE, the biodegradable mulches provide adequate soil temperature and water volume content (WVC), as well as good results in strawberry crops productivity and quality. Biodegradable mulch film seems to be a promising option for the replacement of conventional PE films for strawberry crop production [121].

Physical-mechanical, thermal, and biodegradation performance of air-laid nonwoven fabric of 250 g m^−2^ mass per unit area, aligned and randomly aligned flax fibre composites were investigated. The flax fibres were 22 µm in diameter, with a density of 1.54 g cm^−3^, and fineness of 6.6 dtex. The PLA fibres were 28 µm in diameter, with a density of 1.24 g cm^−3^ and fineness of 3.3 dtex. The aligned composite was made from six layers of flax/PLA yarns compressed and consolidated using a hot compression moulding process, while a randomly aligned composite was made by moulding two nonwoven fabrics with the same parameters as for the aligned composite (pressure, time, and temperature). The rate of biodegradation of randomly aligned flax/PLA structures was found considerably lower than aligned flax/PLA composites. The 120 days after the soil burial test, the randomly aligned flax/PLA composite lost 19% of mass while aligned one lost 27%. The residual flexural strength for randomly aligned composite was reduced by 57% and 80% for aligned composite, while flexural modulus decreased by 50% for randomly and 80% for aligned composite [122].

In the research of Fang C. et al., PLA films and jute nonwoven fabrics were made firstly, then biodegradable composite materials were produced combining PLA films as the matrix with jute nonwoven fabrics as reinforcement using the film-stacking method. PLA, pure jute fibres, and their composites, all show two steps degradation procedures. In accordance with the water evaporation at 50–200 °C, the degradation of cellulose occurs between 250 °C and 400 °C, the jute fibres degraded at about 248 °C, while PLA degraded at 340 °C. With the addition of the jute fibres, the weight loss rate of the composites decreases considerably slower than PLA at 400 °C. Different production conditions have no great influence on the thermal performance of composite materials. Biodegradable composite materials have great potential applications in various fields with advantages of low cost, easy manufacture, low density, and excellent mechanical properties [123].

Comparing the price and recycle cost of polyethylene plastics and the price of biodegradable mulches, it is obvious that biodegradable mulches are currently more expensive [80]. In addition, a study was conducted that used the dichotomous questions to survey consumers’ willingness to pay a higher price for strawberries grown on biodegradable mulch compared to those grown on conventional plastic film. The study reveals that only 10.3% of consumers are willing to pay more for food cultivated with biodegradable mulch, thus supporting the usage of eco-friendly agrotextiles [124]. Taking all the above into account, there is still a long way to develop and produce cost-effective bio-based agrotextiles, as well as increase consumer awareness of sustainable agriculture.

## 5. Recycled Mulches

Although waste management has significantly improved in the EU countries, the European economy currently still losing a considerable amount of potential secondary raw materials, such as metals, wood, glass, paper, etc. In 2010, total waste production in the EU amounted to 2.5 billion tonnes. A limited share of total waste production was recycled (36%), while the rest, that could be recycled or reused, was landfilled or incinerated [125]. In 2012, around 1.3 million tonnes of agroplastic waste in the world was generated in agriculture, where only 55% was entered into the recycling system. Only a few countries have mastered the difficult problem of collecting and processing polymeric materials used in plant and animal production, as well as in the form of packaging. In January 2018, the European Commission announced a new strategy for plastics, which aims, by 2030, to achieve polymer materials recycling or re-use in the amount of 60%. By 2040, the goal is to achieve 100% [113].

At present, 674,000 tons of products are manufactured each year to cover the huge demand for agricultural practice, leaving behind about 1 million tons of waste [112]. The use of existing materials in agriculture, especially natural waste materials, instead of producing new ones, is more cost-effective from an economic point of view. For this and other reasons, textile waste is of great interest. Textile waste primarily means polysaccharide fibres, such as cotton, flax, hemp, jute, and wool as protein fibre [85].

Reducing the amount of nonwoven fabrics by recycling is a significant method of increasing product sustainability. Common materials for conventional nonwovens, such as PE and PET, are easily processed through municipal sorting and recycling streams and redirected to the production of new products [2].

The environmental and economic issues, associated with recycling polymer wastes, have led to a strategy for creating agricultural materials from renewable sources that disintegrate quickly and have no negative impact on the environment. Biological recycling of biodegradable polymers is an important way of reducing plastic waste in the environment, but recycling production wastes and post-consumer items have a prevalent practice [126].

There are two forms of recycling, chemical, and physical recycling. Chemical recycling converts high molecular weight polymers into low molecular weight compounds through chemical processes. The obtained compounds can be reactants in the production of various chemicals and polymers. By physical recycling, manufacturing wastes and post-consumer products via reclamation or commingled plastic waste are reprocessed into new products. Physical recycling is preferable to chemical recycling since it is simpler, less expensive, and more environmentally beneficial [127].

Abidi et al. study the impact of accelerated weather conditions on mechanical, thermal, and physicochemical properties of recycled nonwovens. Research was conducted on recycled nonwoven fabrics of mass per unit area in a range from 200 g m^−2^ to 700 g m^−2^ in order to understand the mechanism of decomposition under the influence of UV light, moisture, and heat. Mass per unit area of cotton waste nonwoven (CWNW) was 200 g m^−2^, with an average thickness of 2 mm, while the mass of the textile waste felt was 500 g m^−2^ (TWF 500) and 700 g m^−2^ (TWF 700), with thickness of 6 and 8 mm. Q-Lab Ultra-Violet/Spray accelerated weathering tester was used according to ASTM D4355 standard. The mass as well mechanical properties of recycled nonwoven fabrics decreased but showed good resistance to accelerated weather conditions. The cotton waste nonwoven lost more than 65% of breaking force after the first month and more than 85% after three months. In the same period of testing the textile waste felt showed more resistance to accelerated weathering conditions and lost around 50% of breaking force. The decrease in breaking elongation of the listed materials after three months of exposure was higher in the machine direction (MD) (CWNW (64.27%) > TWF 500 (62.2%) > TWF 700 (38.77%)) than in the cross machine direction (CD) (TWF 700 (37%) > TWF 500 (18%) > CWNW (7.5%)). The different structures showed good resistance to UV and humidity cycles after three months of study, making recycled mulches an alternative to the plastic films currently in use. Furthermore, a thick layer of mulch could retain soil moisture and at the same time heat the soil, which would bring benefit to plants planted in the early months of the season. The problem of disposing recycled mulches after use due to the resistance to degradation remains [128].

A degradation test on three mulching films, blends of 84% textile industry waste fibres (ramie and cotton fibres with a weight ratio of 4:1) and 16% PVA (poly (vinyl alcohol), PA (polyacrylate), and ST (starch) polymers was performed in research by Tan et al. Ramie fibres were 2–5 cm long, 30 µm in diameter, density of 1.49 g cm^−3^ and with moisture content of 6.5%. The cotton fibres were 13 mm long, 20 µm in diameter, a density of 1.58 g cm^−3^, and with 7.2% of moisture content. The fibre/starch, fibre/PVA and fibre/PVA films had an approximate thickness of 0.35 mm and weight of 40 g m^−2^. Initial tensile strength in machine direction was 2.92 MPa, 3.58 MPa, and 3.96 MPa, respectively, elongation was 27.77%, 23.71%, and 10.92%. Tensile strength in cross-machine direction was 3.52 MPa, 2.62 MPa, and 3.74 MPa and elongation was 9.06%, 10.92%, and 18.31%. The fibre/starch mulches can be used for crop mulching with a shorter growing season, as proved in laboratory and field-testing in different seasons. Degradation in soil were in 2–3 months for fibre/PVA and 3–4 months for fibre/PA films. The materials tested in this study are only suitable for crops with shorter growth times due to the rapid rate of mulch degradation [129].

The experimental data in a study by Briassoulis et al., show that, apart from mulching films, the exposure of various agricultural plastic waste to sun radiation under normal field conditions for the corresponding typical periods of exposure does not cause severe degradation, meaning the plastic waste is not recyclable. The soil contamination by plastic waste throughout their use in the fields varies a lot depending on plastic product category. Mulching films have the greatest fluctuation in foreign material content; therefore, farmers and waste collectors should use unique handling techniques for this type of agricultural plastic waste [130].

The performance of the recyclable wool and flax needle-punched nonwoven fabrics blends, with mass per unit area 100 g m^−2^, 105 g m^−2^, and 110 g m^−2^ were compared by a control sample made of synthetic material—PBSA. The cellulose-based samples were made by flax fibres (average diameter of 20–35 µm and length of 35–40 mm), while the protein-based samples were made from wool fibres (average diameter of 25 µm and length of 30–40 mm). In this study, both types of fibres were equally blended (i.e., wool and flax) [131].

Degradable mulches reduced weed germination by 50% and showed 54.6% lower light transmission compared to the control sample. Mulch based on natural fibres, after 90 days of exposure in the soil, was completely degraded. The recycled natural fibre-based mulches can be used as alternative biodegradable agricultural cover [131].

In a study by Liu et al., needle punched nonwoven agrotextiles produced from recycled wool and flax fibres in various fibre ratios, improved agricultural yield. The recyclable wool (diameter 10–30 µm, length 10–40 mm) and flax fibres (diameter 15–45 µm, length 10–45 mm) were not chemically treated. In addition, low-quality fibres were obtained from spinning mills. Mulches produced from wool and flax fibres of 150, 160, 170, and 180 g m^−2^ were blended at a ratio of 50:50, respectively; mulch of 180 g m^−2^ mass per unit was produced at a ratio of 65:35 (wool: flax). The thickness was in the range of 1.72 to 2.41 mm. The nonwoven mulch of 160 g m^−2^ made from wool and flax fibre blend in ratio 50:50, had a lower cotton yield than commercially on the market available transparent polyethylene film mulch, but higher than commercially on the market available white degradable mulch. New recycled nonwoven mulch did not degrade as quickly as the white degradable mulch, because of different degradable mechanisms during the cropping period, confirming that the newly developed nonwoven mulch is suitable for agricultural performance in an arid continental climate area [132].

The possibility of recycling agrotextile wastes itself, especially mulches, is greatly encouraged. However, due to challenges in management and technological elements during collecting, sorting, recycling, and purification of recovered materials, textile waste has not yet been fully utilised [133]. The low amount of plastic mulches is recycled due to the high level of contamination. Polyethylene mulches used in vegetable production are too soiled and polluted to be recycled directly from the field. Contamination by pesticides, fertilizers, dirt and debris, moist vegetation, silage juice water, and UV additives can make up to 40–50% of agricultural plastics weight, notably in mulch films and drip irrigation tape. Plastic films that contain more than 5% contaminants by weight will not be recycled [67].

## 6. Conclusions

Nonwoven mulches made from natural fibres are completely biodegradable over a period of 3–6 months, which makes them the best choice for annuals and seedlings with shorter growing periods. The research on mulches made of natural fibres, which meet the mulching requirements, are mostly produced from flax, hemp, jute, linen, wool, and cotton fibres, by mechanical process on card and bonded by needling process. The mulches are produced in mass per unit area in range from 180 to 400 g m^−2^ and thickness from 0.29 to 0.63 mm. Additionally, raw materials for those mulches are widely available, nonwoven fabrics from natural fibres is easy to produce, mulches are harmless to the environment and after decomposition could be used as a fertilizer.

Modifying cellulose or producing blends of natural fibres and biopolymers and/or man-made fibres prolongs the life of mulches but requires adequate care after use. Biodegradation of biopolymers requires certain conditions (abiotic and biotic factors, temperatures, etc.) and must degrade for 90% in 180 days, according to applicable standards for biopolymers. Research on mulches made from biopolymers is mostly related to spunbond or meltblown nonwoven mulches made of PLA and their blends with PHA or PBS polymers. Research showed that polylactic fibres are an alternative to PP, PE, or PET fibres because of similar properties to non-biodegradable synthetic fibres (controlled crimp, smooth surface, and low moisture absorption) while PLA fibre has better weather resistance and flame retardancy relative to PET fibres, as well as rapid degradation in activated sludge. The mass per unit area of those mulches produced by spunbond or spunmelt technology is in the range of 50 to 100 g m^−2^ and thickness ranged from 593 to 666 μm, i.e., with a structure similar to conventional mulching foil compared to mechanically produced nonwovens. Biopolymer mulches are good alternative solutions for perennial plants. It needs to be emphasized that most of the studies on the suitability of biopolymers for agrotextile production provide just part of the information. Specifically, studies lack key information and data, such as raw material composition, biodegradation process monitoring, soil analysis to prove complete biodegradation including the impact of biopolymer on soil health, as well as its influence on plants.

Biodegradation of recycled mulches largely depends on their composition, but with adequate municipal sorting and recycling streams can be reused for both annual and perennial plants with the greatest economic advantage. The technology of nonwoven mulches production from recycled fibres is similar to the production of mulches made of natural fibres, i.e., mechanically on a card, bonded by needling. Therefore, the mulches masses per unit area are high, usually from 150 g m^−2^ up to 700 g m^−2^ with a corresponding thickness of 1.7 to 8.0 mm. The most common recycled waste for mulches production is from cotton, wool, and flax waste and generally a mixture of textile waste from factories.

The application and end-use of the agrotextile product will determine suitable raw material, where, in terms of biodegradation and ecological footprint, natural fibres are the best choice.

Nowadays, it is imperative to have or use environmentally sustainable products, where due to this trend, many products are declared as biodegradable. Limited biodegradation rates raise fundamental questions about whether materials classified as “biodegradable” are truly biodegradable or can only be composted. It must be emphasized that biodegradable plastics (made from renewable raw materials) are not automatically biodegradable, while on the other hand biodegradable plastics are not necessarily made from renewable raw materials.

## Figures and Tables

**Figure 1 polymers-14-02272-f001:**
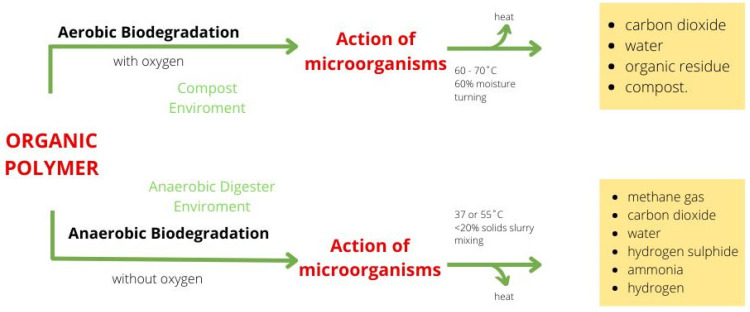
Aerobic and anaerobic degradation pathways for organic polymers [30].

**Figure 2 polymers-14-02272-f002:**
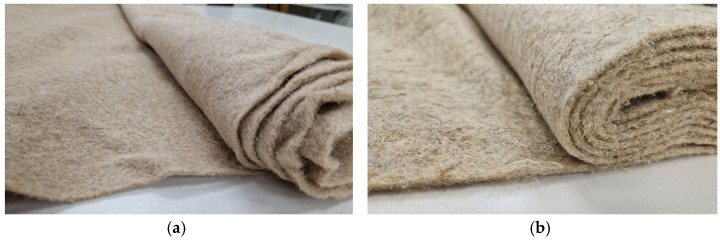
The nonwoven mulches produced by (**a**) jute and (**b**) hemp fibres.

**Figure 3 polymers-14-02272-f003:**
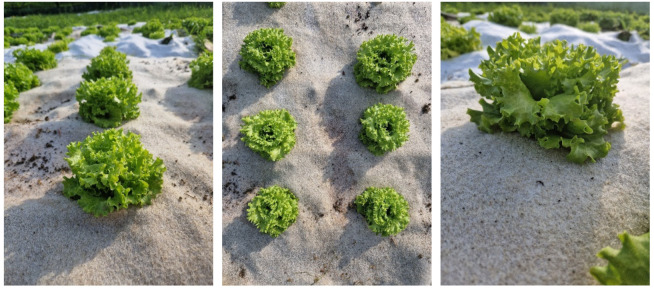
Protection from weed infestation by nonwoven mulches.

**Figure 4 polymers-14-02272-f004:**
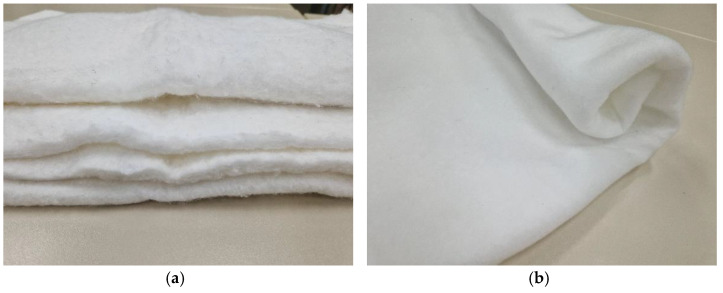
The nonwoven mulches produced by (**a**) viscose and (**b**) PLA nonwoven mulches.

**Table 1 polymers-14-02272-t001:** Standards related to the biodegradability of polymers in soil environment [25].

Measuring Method	Standard	
O_2_ consumption or CO_2_ production	OECD 304A; 1981	Inherent biodegradability in soil
	ISO 11266:1994	Soil quality—Guidance on laboratory testing for biodegradation of organic chemicals in soil under aerobic conditions
	ASTM D5988-96	Standard test method for determining aerobic biodegradation in soil of plastic materials or residual plastic materials after composting
	ISO 17556:2003	Plastics—Determination of the ultimate aerobic biodegradability in soil by measuring the oxygen demand in a respirometer or the amount of carbon dioxide evolved

**Table 2 polymers-14-02272-t002:** Natural fibres physical and chemical properties with biodegradation time [42,49,51,52,53,54,55,56,57,58,59,60,61].

Fibre	Diameter (μm)	Length (mm)	Cellulose (wt.%)	Hemi Celluloses (wt.%)	Lignin (wt.%)	Pectin (wt.%)	Waxes (wt.%)	Degradation Time (Months)
Abaca	10–30	4.6–5.2	45.4	38.5	14.9	1.1	2.0	
Alfa	14–17	1–2	37.0	21.0	22.0	10.0	-	
Bagasse	10–34	0.8–2.8	62.5	12.5	7.5	4.0	-	
Banana	12–30	0.4–0.9	34.5	20.5	26.0	-	-	
Bamboo stick	25–88	1.5–4.0	46.0	0.3	45.0	4.0	-	12–36
Coir	7–30	0.3–3.0	89.0	4.0	0.8	6.0	0.6	
Cotton	12–35	15–56	82.7	5.7	0.0		0.6	
Coconut	100–450	0.3–1	70.5	16.5	2.5	0.9	-	1–6
Flax - single fibres - technical fibres	12–37 40–620	15–20 500–750	62.0–72.0	18.6–20.6	2.0–10.0	0.9	1.7	3
Hemp - single fibres - technical fibres	16–50 40–620	10–15 700–1500	64.0–78.3	16.0–22.4	2.9–5.7	-	0.8	3–8
Henequen	150–250	700–800	67.0	16.0	9.0	0.2	0.5	
Jute - single fibres - technical fibres	18–20 -	0.8–6.0 2000–3000	56.0–71.5	13.6–35.0	2.9–5.7	-	0.5	6–18
Kapok	20–43	10–35	53.5	21.0	17.0	2.0	-	
Kenaf	12–36	1.4–11	80.5	17.5	8.3	4.0	-	6–12
Pineapple	8–41	3–8	72.0	14.0	0.8	2.0	-	
Ramie - single fibres - technical fibres	50 5000	60–250 ~2000	60.0	11.5	8.0	1.2	-	
Sisal	100–300	600–1500	62.5	21.0	12.0	0.8	3.0	12

**Table 3 polymers-14-02272-t003:** Characterization and degradation rate of biopolymers [47,101,102].

Biopolymer	Source	Properties	Composting Time, Days	Degradation Time, Months
Polylactide (PLA)	Sugar beet, corn starch, sugar cane	Hydrophobic, compatibility with polyesters, low moisture absorption rate, resistance to UV radiation, low thermal stability (60 °C), compostable improves the stiffness of textiles, low crystallization pace, eco-friendly.	45–60	20
Polyhidroxyalkanoates (PHA)	Stored in bacterial cells as reserve material	Good mechanical properties, suitable as biomass material, easy to process, tough and durable, UV-resistant, water-resistant, eco-friendly.	-	12
Poly (ε-caprolactan) (PCL)	Petrol	Low UV resistance, low melting temperature (60 °C) high elasticity, eco-friendly.	6–28	-
Thermoplastic starch blends	Potato, corn, wheat, rice, mixing with bio-based polymers (PLA, PCL)	Hydrophobic character, low permeability to water, better mechanical properties, properties dependent on composition, low cost, global accessibility eco-friendly.	45–56	-
Poly(hydroxylbutyrate) (PHB)	Produced and stored by bacteria	Stiff and brittle, high crystallinity, difficult to process, weak impact resistance, can degrade via hydrolysis at high temperatures, low chemical resistance, suitable for modification.	21–28	6–10
Poly(butylenesuccinate) (PBS)	Polycondensation reaction of 1, 4-butanediol with succinic acid	Flexible, excellent impact strength, chemical and thermal resistance, composite material, comparable mechanical properties to PE and PP, processing capacity, eco-friendly.	-	-
Cellulose acetate (CA)	Acetylation from wood pulp by acetic acid or its anhydride to cellulose	Medium mechanical properties, ductile, scratch and scrub resistant, antistatic, good insulation, oil-resistant, high processing efficiency.	-	-
Viscose (CV)	Wood pulp	Regeneration of cellulose fibres from solutions of derivatives (e.g., viscose, modal fibres) or by regeneration of cellulose fibres from solutions of cellulose (e.g., cuprammonium).	60	-

## Data Availability

Not applicable.

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
