# Peer review of "Biodegradable Nonwoven Agrotextile and Films—A Review"

_polymers, 2022, doi:10.3390/polym14112272_

Round 1
Reviewer 1 Report
At the moment, the review by Dragana Kopitar et al is not ready for publication, although the text of the review is easy to read and presented in an accessible language (i.e., it will be understandable even to a non-specialist). The authors need to make a number of corrections and clarifications to the review. Thus, in the work, instead of nonwoven materials, the authors often consider films that are not related to the topic of the review. If the authors have input films, it is advisable to change the title of the review. The paper does not consider the mechanical characteristics of materials, and without them it is difficult to compare the described materials and evaluate their applicability. The review does not take into account the structural features of materials, the fibers from which they are made, etc. It is not clear what kind of flax fibers are used (long or short, containing shives or not), etc.
In my opinion, questions regarding the degradation of polymers can be presented in a more concise form. Table 2 shows the biodegradation time for a number of natural fibers. However, the question arises why flax fibers biodegrade faster than, for example, bamboo? Although the content of cellulose in flax is higher. For hemp, the gerometric dimensions can be taken from other sources. Also confusing is the cellulose content of hemp fibers, which needs to be checked.
Table 4 Viscose is not a polymer. Also, fibers obtained from solutions of cellulose in NMMO do not belong to viscose. The properties of hydrated cellulose fibers are very dependent on the characteristics of the original cellulose, the method of fiber formation, etc., the authors do not discuss this.
In conclusions, I recommend the authors to focus on the polymer part (synthesis, production, structure, properties, etc.), and not on materials science.
Author Response
I would like to thanks the reviewer who took the time to review the paper.
- Thus, in the work, instead of nonwoven materials, the authors often consider films that are not related to the topic of the review. If the authors have input films, it is advisable to change the title of the review.
Title is changed
- The paper does not consider the mechanical characteristics of materials, and without them it is difficult to compare the described materials and evaluate their applicability. The review does not take into account the structural features of materials, the fibers from which they are made, etc. It is not clear what kind of flax fibers are used (long or short, containing shives or not), etc.
All reference were checked up for mechanical and structural characteristics of materials as well fibres. From each reference, where data was available, information’s were incorporated in the manuscript. It should be point out that most of the researches doesn’t provide those information. Just some of them specify the exact mechanical properties of materials. Often there are just comments on the trend of decreasing tensile properties, which are indicators of changes caused by biodegradation in mechanical properties. For example, after 21 days after burial, the breaking force of the PLA spunbond nonwoven fabric had decreased by around 45 percent, indicating that they were gradually biodegrading. Both breaking force and elongation remained constant for PP, indicating that the fabric did not deteriorate much over 140 days [76]. Similar conclusions can be seen in another study where tensile strength of flax fabrics produced by pre-needling and hydroentanglement after the two months of field trial exposure, was reduced by around 30% compared to the as-produced fabrics which had a normalized strength of 0.013 N/gsm and increased more than 50 fold after hydroentanglement. The bursting strength results also showed a reduction after exposure [74].
- In my opinion, questions regarding the degradation of polymers can be presented in a more concise form.
Text regard to biodegradation is rearranged in order to present it in form that is more concise.
- Table 2 shows the biodegradation time for a number of natural fibers. However, the question arises why flax fibers biodegrade faster than, for example, bamboo? Although the content of cellulose in flax is higher. For hemp, the gerometric dimensions can be taken from other sources. Also confusing is the cellulose content of hemp fibers, which needs to be checked.
Table 2 is corrected, information related to hemp fibres are checked and corrected. After table 2 explanation of biodegradation time considering composition of cellulose fibres is given.
- Table 4 Viscose is not a polymer. Also, fibers obtained from solutions of cellulose in NMMO do not belong to viscose. The properties of hydrated cellulose fibers are very dependent on the characteristics of the original cellulose, the method of fiber formation, etc., the authors do not discuss this.
Thank you for your comment. Viscose is not biopolymer so the title of the chapter as well title of the Table 4 is changed. After Table 4, at the end of the chapter authors discuss the method of fibre formation.
- In conclusions, I recommend the authors to focus on the polymer part (synthesis, production, structure, properties, etc.), and not on materials science.
In conclusion part synthesis, production, structure, properties of mulches made of natural fibres, biopolymers as well as recycled mulches is added.
Reviewer 2 Report
- Avoid using “we” or “I” in your text, such as done in the abstract.
- There is only one figure in this paper. It is recommended to add more figures for better understanding the presented information. For example, images from different fibers or agro-textiles may be useful.
- It is recommended to add a table or figure presenting the international trend of the production and consumption of agro-textiles and related products in recent years.
Author Response
I would like to thanks the reviewer who took the time to review the paper.
- Avoid using “we” or “I” in your text, such as done in the abstract.
Thank you for the comment, abstract is corrected.
- There is only one figure in this paper. It is recommended to add more figures for better understanding the presented information. For example, images from different fibers or agro-textiles may be useful.
Images of different nonwoven agrotextile and mulching application is given.
- It is recommended to add a table or figure presenting the international trend of the production and consumption of agro-textiles and related products in recent years.
Additional data of global agrotextiles market size is added in introduction part. Unfortunately, figures and detailed information about production and consumption of agrotextiles in recent years is not available for public. Only the market analyses that must be paid (around 5000 dollars)
Round 2
Reviewer 1 Report
The authors made the appropriate corrections to the review and took into account the comments of the reviewer. The work may be reviewed further by the editor. Below are additional recommendations:
Table 2. I recommend that the authors indicate the ranges of lengths (diameters) of fibers of natural fibers (for example, for flax, the length of the fibers can be 5 mm or more - https://doi.org/10.3390/fib10050045 ). Variations in length are very important (focusing on the length of the fiber one or another method of obtaining a nonwoven or another type of material is chosen), while the diameters vary in a smaller range and are often close when moving from one type of fiber to another.
Lines 248, 249. "Bamboo fibres, for example, take up to 36 months to degrade and have a lower cellulose content than flax fibres, which degrade within 3 months" - I recommend that the authors recheck the timing of fiber degradation. In this case, it is important to understand which fibers are described in the cited papers. Depending on the history of obtaining bamboo and linen fibers, the degradation period may vary. Plant age matters. In a number of works, it is noted that for bamboo, with age, a decrease in the content of cellulose is observed and the content of lignin and hemicellulose increases.
Author Response
I thank the reviewer for the comments. Following corrections are done.
- Table 2. I recommend that the authors indicate the ranges of lengths (diameters) of fibers of natural fibers (for example, for flax, the length of the fibers can be 5 mm or more - https://doi.org/10.3390/fib10050045 ). Variations in length are very important (focusing on the length of the fiber one or another method of obtaining a nonwoven or another type of material is chosen), while the diameters vary in a smaller range and are often close when moving from one type of fiber to another.
In Table 2 diameters and lengths of natural fibres are corrected (marked in green).
- Lines 248, 249. "Bamboo fibres, for example, take up to 36 months to degrade and have a lower cellulose content than flax fibres, which degrade within 3 months" - I recommend that the authors recheck the timing of fiber degradation. In this case, it is important to understand which fibers are described in the cited papers. Depending on the history of obtaining bamboo and linen fibers, the degradation period may vary. Plant age matters. In a number of works, it is noted that for bamboo, with age, a decrease in the content of cellulose is observed and the content of lignin and hemicellulose increases.
The text about bamboo fibres in manuscript is corrected and complement (marked in green).